# Assessment of Dietary Folate Intake and Pill Burden among Saudi Patients on Maintenance Hemodialysis

**DOI:** 10.3390/ijerph182312710

**Published:** 2021-12-02

**Authors:** Ibrahim Sales, Ghada Bawazeer, Ahmad R. Tarakji, Feriel K. Ben Salha, Nourah H. Al-Deaiji, Marwah Saeed, Rawan S. Idris, Mohammad H. Aljawadi, Majidah A. Aljohani, Mansour Adam Mahmoud, Wajid Syed

**Affiliations:** 1Department of Clinical Pharmacy, College of Pharmacy, King Saud University, Riyadh 11451, Saudi Arabia; gbawazeer@ksu.edu.sa (G.B.); maljawadi@KSU.EDU.SA (M.H.A.); wali@ksu.edu.sa (W.S.); 2Consultant Nephrologist, St. George Medical Centre, Kitchener, ON N2R 0H3, Canada; dartmo@yahoo.com; 3Primary Point of Contact, Roche Tunisia & Libya, Roche Tunisie SA, Les Arcades, Lac Loch Ness Street, Berges du Lac, Tunis 1053, Tunisia; F.bensalha@outlook.com; 4Pharmaceutical Care Division, National Care Hospital, Riyadh 11461, Saudi Arabia; n.alduaiji90@gmail.com; 5Pharmaceutical Care Division, King Faisal Specialist Hospital & Research Centre, Riyadh 11564, Saudi Arabia; marwahsaeed@hotmail.com; 6Pharmaceutical Care Division, Green Crescent Hospital, Riyadh 12711, Saudi Arabia; R.idris@hotmail.com; 7Pharmaceutical Care Division, King Saud Medical City, Riyadh 12746, Saudi Arabia; majda.a.aljuhani@gmail.com; 8Department of Clinical and Hospital Pharmacy, College of Pharmacy, Taibah University, Al-Madinah Al-Munawarah 42353, Saudi Arabia; mammm.99@gmail.com

**Keywords:** folic acid, anemia, burden of oral therapy, maintenance hemodialysis

## Abstract

The aim of this study was to assess the adequacy of dietary folate intake and perceptions of pill burden among Saudi patients on maintenance hemodialysis (MHD). This was a cross-sectional study of adults (>18 years) on MHD (>3 months) attending the dialysis unit at King Saud University Medical City. Patient demographics, dietary folic acid intake, and perceptions of pill burden were collected. Fifty-four patients met the eligibility criteria, with a mean age of 57 ± 15.5 years. The majority were females (63%), and the most prevalent comorbidities were diabetes (43%) and hypertension (76%). The average number of medications/patients was 11 ± 2.9, and most patients were receiving folate supplementation (68.5%). The average dietary folate intake was 823 ± 530 mcg/day. Pill burden was bothersome, primarily due to taking too many medications (57%) while taking medications at the workplace was the least bothersome burden (17%). The reported high pill burden and adequate dietary folate intake by Saudi patients on MHD indicates that the omission of folate supplementation may be advantageous for this special population.

## 1. Introduction

Anemia is one of the most common complications of chronic kidney disease (CKD), with iron-deficiency anemia being the most common manifestation besides a decrease in erythropoietin activity [1]. When indicated, the use of human recombinant erythropoietin (rHuEPO) and iron supplementation is usually sufficient to correct anemia and is considered the standard of care [2,3]. Anemia due to folic acid and B12 deficiency is uncommon in CKD patients. Although these water-soluble vitamins are highly removed during dialysis, regular supplementation is controversial [4,5,6,7]. Maintaining a normal, folate-balanced diet is reasonably sufficient to replace folate loss in the dialysate [8]. Nonetheless, observational data suggest that malnutrition is also common among CKD patients [9]. Folate supplementation has been associated with an enhanced effect of erythropoietin treatment in patients with folic acid deficiency [10,11,12]; however, KIDIGO guidelines do not advocate the use of adjuvants (such as folic acid) except in instances of deficiency or inefficacy [3]. Most dialysis patients receive adequate folate supplementation from folate-fortified foods in countries that require mandatory fortification [6]; however, due to the fact that folic acid has relatively low toxicity, routine supplementation with folic acid in CKD patients on dialysis is still a routine practice in most dialysis units as a preventative measure [2,13,14]. In the Kingdom of Saudi Arabia (KSA), food fortification is encouraged but not mandatory [15]. According to data compiled by the Food Fortification Initiative, wheat fortification in KSA is voluntary, and no legislation has been passed regarding maize and rice (the main staple of a typical diet in the country).

Another aspect that can affect patient adherence and quality of life is the daily pill burden in patients who are on maintenance hemodialysis (MHD). CKD has been associated with the highest pill burden compared to any other chronic disease state [13,16,17]. The complexity of the medication regimens of MHD patients contributes to an increase in the rate of non-adherence and negatively affects their quality of life. Studies have estimated that the mean daily pill burden in chronic hemodialysis (HD) patients is around 10–12 medications per day [14]. Phosphate binders have been implicated as the most common drug associated with non-adherence in MHD patients [18]. The aim of this study was to assess the adequacy of dietary folate intake and the perceptions of pill burden among Saudi patients on MHD.

## 2. Materials and Methods

### 2.1. Study Design and Setting

A cross-sectional study was conducted in the dialysis unit at King Saud University Medical City (KSUMC) in Riyadh. KSUMC is a tertiary teaching hospital that serves the employees of King Saud University, their dependents, and Saudi nationals not affiliated with the university. The majority of patients are native Arabic speakers. The study period ran from October–November 2015 and was concluded in January 2016.

### 2.2. Participants and Enrollment

All adult patients (≥18 years) from both genders attending the KSUMC HD unit who had been on chronic HD for at least three months prior to the study and were able to communicate through the questionnaire administration were included. Patients were excluded if they were on peritoneal dialysis, were pregnant, had psychiatric illnesses, or were under palliative care.

### 2.3. Folic Acid and Pill Burden Assessment Tools

Two surveys were adopted and administered to the patients. The folate-rich food survey [19,20] was used to measure daily dietary intake of folic acid. This survey was adopted and modified based upon the World Health Organization (WHO) estimates of foods common to the Saudi diet and the Saudi Healthy Food Palm [21,22]. The perceived burden of the oral therapy survey [23] assessed the complexity of oral therapy.

### 2.4. Data Collection

The administration of the two questionnaires to consenting patients was conducted by four authors (F.B., N.S., M.S., and R.I.). Prior to the data-collecting period, pilot questionnaires were administered to five patients and then modified, as required, to ensure optimum understanding by the patients. During this phase, the researchers observed each other to standardize the questionnaire administration practice and minimize variations in data collection and interpretation of patients’ responses. All subsequent questionnaire administration was performed independently. Eligible patients were approached to obtain their consent at the beginning of their scheduled MHD session. Both questionnaires were administered during a single MHD session, and the researchers spent approximately 20–30 min with each participant. Researchers began with the food frequency questionnaire and all diet-related questions first, and then the pill burden questionnaire was completed by the same researcher. The researchers asked the participants each survey question and recorded their responses on the data collection forms.

### 2.5. Data Analysis

The content of dietary folic acid intake was adjusted and calculated for each patient and presented as the amount of folic acid intake per day (amount in mcg/day). The perceived burden of therapy (PBOT) survey consisted of 6 items that measured how much patients were bothered by the number of medications prescribed, the medication size, adverse effects, the number of times treatment should be administered during the day, the need to take medications at work or in social contexts, and the need to drink to take their medications. These items are answered using a 5-point Likert Scale (1—Not bothered at all to 5—Extremely bothered). The total score of PBOT was 30 (a maximum score of 5 for each question); therefore, a cutoff score of 15 was used. Patients who scored 15 and above were considered to have a high level of PBOT, and patients who scored less than 15 were considered to have a low level of PBOT.

### 2.6. Ethical Considerations

This study was approved by the KSUMC Institutional Review Board Project No. E-19-3858.

### 2.7. Statistical Analysis

Continuous variables were presented as means and standard deviations. Categorical variables were presented as numbers and percentages. Mann–Whitney tests were used to test for differences between the continuous variables because the data was not normally distributed. Chi-square tests or Fisher’s exact tests were used to test for differences among the categorical variables. A *p* value < 0.05 was considered significant. The Statistical Package for Social Science (SPSS) Version 22 (IBM Corp., Armonk, NY, USA) was used for all statistical analyses.

## 3. Results

Over a 3-month period, a total of 63 patients on MHD were screened, and 54 patients met the eligibility criteria. A total of nine patients were excluded (Figure 1). Most of the patients attended three sessions per week (92.6%), while a minority attended four sessions per week (7.4%). The mean patient age was 57.2 ± 15.5 years, the majority were female (63%), and 74% were married. The mean dialysis vintage was 6 ± 5.5 years. The baseline patient demographics and clinical characteristics are summarized in Table 1.

Fifty-three (98.1%) patients had at least one active disease. The patients were suffering from comorbidities, including hypertension (75.9%), anemia (48.1%), diabetes mellitus (42.6%), ischemic heart disease (IHD) (13%), hypothyroidism (11.1%), heart failure (HF) (7.4%), atrial fibrillation (AF) (5.6%), myocardial infarction (MI) (3.7%), and systemic lupus erythematosus (SLE) (3.7%). A failed renal transplant was reported in 13% of the patients. 

The average number of medications was 11 ± 29. The most common medication classes prescribed were phosphate binders (90.7%), vitamin D analogues (83.3%), multivitamins (75.9%), erythropoiesis-stimulating agents (ESAs) (72.2%), iron supplements (68.5%), folic acid (68.5%), statins (64.8%), proton pump inhibitors (61.1%), non-steroidal anti-inflammatory drugs (NSAIDS) (51.9%), calcium channel blockers (33.3%), beta blockers (29.6%), insulin (25.9%), cholecalciferol (22.2%), anticoagulants (20%), antibiotics (18.5%), and oral antihyperglycemic medications (7.4%). In addition, the average number of pills per day was 16 ± 3.6.

### 3.1. Folic Acid Intake

Data from the folate-rich food questionnaire indicated that the average amount of dietary folic acid intake was 822.9 ± 530.7 mcg/day. Moreover, 37 patients (68.5%) were receiving additional folate supplementation. Those on folic acid supplementation were most commonly females (67.6%), non-smokers (78.4%), married (73%), and had a formal education (64.9%).

### 3.2. Perceived Burden of Therapy

The PBOT survey showed good reliability (Cronbach’s α = 0.77). The average number of medications per patient was 11.0 ± 3, with an average of 15.21 ± 5.55 pills/patient/day. Table 2 summarizes the patients’ perceptions of PBOT. Patients with low levels of PBOT had a higher mean age compared to patients with high levels of PBOT (*p* = 0.02) (Table 3). 

## 4. Discussion

This was a single-center study designed to assess dietary folic acid consumption and the perceived pill burden in patients undergoing MHD. The results of our study indicated that the majority of participants consumed adequate levels of folic acid from their normal diets. To the best of our knowledge, this is the first study to assess the amount of dietary folic acid intake among patients on MHD. It has been suggested that Saudi Arabia has a high prevalence of folate deficiency; however, to our knowledge, no local studies have been performed to assess the dietary consumption of folic acid.

In accordance with the U.S. Public Health Service, Center for Disease and Control (CDC), and Health Canada recommendations, the recommended daily intake of folic acid for healthy adults is 400 mcg/day [24,25,26,27]. However, in CKD patients, the available guidelines do not mention folic acid as part of the management and recommend against routine supplementation when the signs and symptoms of anemia are absent [3]. Palabiyik et al. compared the folate levels of participants on MHD with subjects without renal insufficiency [28]. The healthy volunteers’ folate levels were significantly higher in the MHD group, and this was attributed to the majority of MHD patients taking folic acid-containing multivitamins. Malyszko et al. reported that 68% of hemodialysis patients being treated for anemia were concomitantly receiving folic acid therapy [29]. Lee et al. measured serum folate, vitamin B12, and red cell folate concentrations by radioimmunoassay in continuous ambulatory peritoneal dialysis and hemodialysis patients not taking folate supplementation [6]. Folate deficiency occurred in 10% of the hemodialysis patients, and no macrocytic anemia was reported in any of the patients. 

Well-nourished patients on MHD are less likely to develop folic acid deficiency as hemodialysis per se is not an indication for folic acid supplementation unless the patient is suffering from hyperhomocysteinemia, hyporesponsiveness to erythropoietin therapy, or malnutrition [3,5,30]. Many factors could contribute to malnutrition in hemodialysis patients such as dialysis duration, low frequency of taking three daily meals, inadequate nutrient intake, and changes in taste. Additionally, the disease itself could cause additional burden and psychological problems that could affect their dietary behaviors [31]. Nevertheless, folic acid concentrations are not considered to be reliable measures of actual folate tissue concentrations. They more accurately reflect recent dietary intake [32]. 

Pill burden is a major concern for both healthcare providers and patients. Complex medication regimens and a high number of medications and/or pills per day could significantly affect medication adherence and may become burdensome for many patients. Compared with other chronic diseases, CKD patients and patients on hemodialysis are typically prescribed an excessive number of medications due to multiple comorbidities and dialysis-related complications [14,33,34]. In the present study, we measured the PBOT from the patients’ perspective. We found that approximately 60% of patients were bothered by taking too many medications. An average of 11.0 ± 3 medications/patient and an average of 16 ± 3.6 pills/patient/day were reported by our patients. This is in agreement with previous findings in the literature indicating that CKD patients take an average of 10–12 medications daily [13,14,35]. 

As found in previous studies, phosphate binders, followed by vitamin D analogues, were the most commonly prescribed medications in our cohort and were the primary reasons for increased levels of dissatisfaction with medication regimens and pill burden. Additionally, phosphate binders have been associated with high levels of medication non-adherence [18,35,36,37], which we did not measure for our population.

On the pill burden survey, approximately 60% of patients were bothered by taking too many medications. Although taking medications at the workplace was the least bothersome aspect, the average age of the participants was close to 60; therefore, many of the participants may have been retired. Over half of the participants were bothered by medication side effects. Adverse effects of medications have also been associated with non-adherence [38,39,40,41,42,43,44]. Ghimire et al. reported that some patients independently decide to “experiment” with their medications by temporarily or permanently discontinuing therapy to determine whether the adverse reaction is due to the medication [39]. The large pill sizes, which are typically associated with phosphate binders, concerned approximately 43% of the patients. Many similar surveys have confirmed this finding. Pill sizes further pose a challenge for this population due to fluid restrictions and difficulty swallowing [39,45,46,47,48,49,50]. One-third of our patients complained about taking some of their medicine(s) multiple times daily. Frequency has likewise been highly associated with deficiencies in medication adherence [46,48,51,52]. 

Promising strategies and interventions to decrease pill burden include an increased involvement of clinical pharmacists, incorporating structured medication reconciliation (SMR), and deprescribing. Mateti et al. investigated the impact of a pharmacist-led enhanced pharmaceutical care program upon the health-related quality of life (HRQoL) and clinical outcomes in MHD patients over a 12-month period [53,54]. Patients randomized to the pharmaceutical care group had significant reductions in interdialytic weight gain and blood pressure as well as significant increases in hemoglobin levels, medication adherence, and six HRQoL domains. Structured medication reconciliation (SMR) requires inter-professional collaboration. The clinical pharmacist plays a critical role in coordinating the process of medication reconciliation. Such services can reduce polypharmacy, medication-related problems, and healthcare-related costs, especially for MHD patients [55]. A quality improvement project aimed at improving patient treatment adherence in patients on MHD in an outpatient hemodialysis unit included multiple initiatives [56]. The project included a comprehensive SMR on regular intervals and an assessment of treatment adherence via both objective measures (i.e., laboratory values, vital signs, and interdialytic weight gain) and the administration of a pre- and post-End-Stage Renal Disease Adherence Questionnaire (ESRD-AQ). The study demonstrated a statistically significant improvement in the accuracy of medication profiles and improvements in objective measures and the use of pharmacy services. Furthermore, McIntyre et al. developed a deprescribing tool for MHD patients for certain types of medication classes [57]. They hypothesized that if deprescribing reduces polypharmacy, it can lead to a decrease in medication burden and possibly increase adherence. After a 6-month period, 57% of study participants had a reduction in pill burden.

Our study had several limitations. The patients were recruited from a single HD center in KSA; therefore, the sample size was small and may affect the results and generalizability of the study. Further, different populations in the same and different countries might have different dietary folic acid intakes. Another limitation is the lack of realization of portion size in most of our Saudi patients and the fact that the majority have no fixed meal schedules. This could possibly make the data collection for the food frequency questionnaire less accurate and more vulnerable to either over- or under-estimation of the actual folic acid intake. Additionally, the folic acid amounts were estimated based on the folate-rich food survey, and some of the amounts were retrieved from the WHO. Furthermore, this study did not take into consideration patient-reported medication adherence. Finally, possible variations in the researchers’ skills and data collection may have existed, although the initial observation period sought to minimize these potential discrepancies. 

## 5. Conclusions

Folate supplementation in centers providing services to Saudi patients on MHD may not be necessary. The dietary folate intake in Saudi patients on MHD appears to be more than adequate. The reported high pill burden by Saudi patients on MHD indicates that the omission of folate supplementation may be advantageous for this special population, already plagued with a high prevalence of polypharmacy and suboptimal medication adherence. These findings need to be confirmed in future, larger, multi-center studies to provide general guidance for the national and international communities. 

## Figures and Tables

**Figure 1 ijerph-18-12710-f001:**
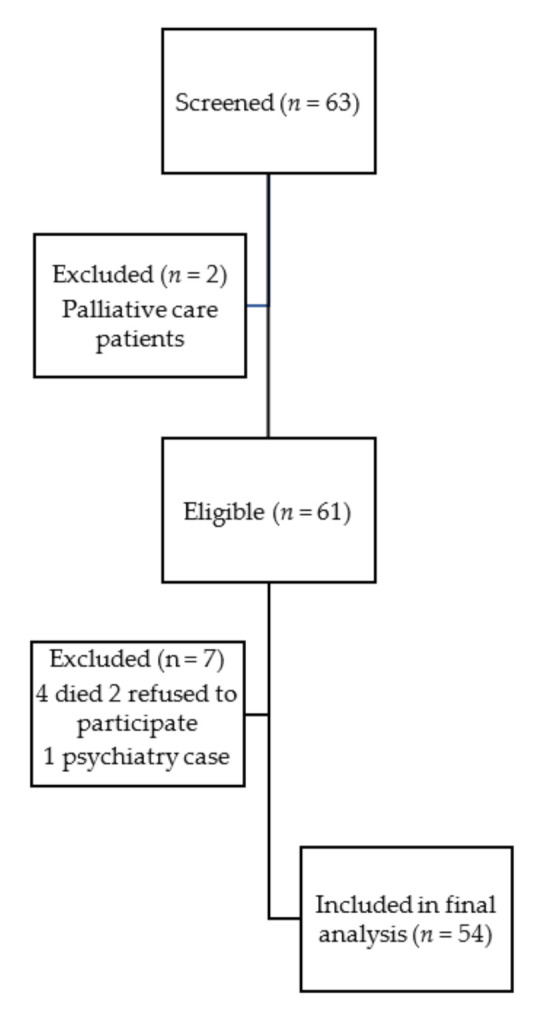
Flowchart of patient recruitment.

**Table 1 ijerph-18-12710-t001:** Patient demographic characteristics.

Characteristics (n = 54)	N (%)
Age (years) *	57.2 ± 15.5
Gender	
Male	20 (37)
Female	34 (63)
Smoking status	
Non-smoker	45 (83.3)
Smoker	9 (16.7)
Marital status	
Married	40 (74.1)
Single	14 (25.9)
Level of education	
Illiterate	22 (40.74)
School education	17 (31.5)
University	15 (27.8)
Hemoglobin level (g/L) *	109.04 ± 21.3
Hct% *	34.5 ± 13
MCV *	87 ± 7.1
Number of medications *	11 ± 2.9
Number of pills/patient/day *	15.21 ± 5.55
Number of comorbidities *	4.6 ± 2.3
Dialyzer type	
Fx60	23 (42.6)
Fx80	31 (57.4)
Number of sessions per week	
Three sessions	50 (92.6)
Four sessions	4 (7.4)
Duration of the session	
Three hours	5 (9.3)
Three and a half hours	20 (37.0)
Four hours	29 (53.7)
Folic Acid supplementation	
Yes	37 (68.5)
No	17 (31.5)
Folic Acid dose	
1 mg	18 (33.3)
5 mg	36 (66.7)

* mean ± SD. Hematocrit (Hct), Mean Corpuscular Volume (MCV).

**Table 2 ijerph-18-12710-t002:** Participant responses to the Perceived Burden of Therapy (PBOT) Survey.

Variable	Not Bothered at All N (%)	Somewhat Bothered N (%)	Moderately Bothered N (%)	Very Bothered N (%)	Extremely Bothered N (%)	Mean Perception Score *
Taking too many medications	19 (35.8)	3 (5.7)	14 (26.4)	7 (13.2)	10 (18.9)	1.7
Side effects of medicines	16 (30.2)	9 (17)	9 (17)	4 (7.4)	15 (28.3)	2
Pill size (too big)	20 (37.7)	10 (18.9)	6 (11.3)	7 (13.2)	10 (18.9)	2.2
Taking medicines many times per day	22 (41.5)	13 (24.5)	8 (15.1)	2 (3.8)	8 (15.1)	1.5
Taking medicine at the workplace	41 (77.4)	3 (5.7)	1 (1.9)	5 (9.4)	3 (5.7)	1
The need to drink to take medicines	37 (69.8)	3 (5.7)	4 (7.5)	6 (11.3)	3 (5.7)	1.6

* The mean score was based upon the 5-point Likert Scale (1—Not bothered at all to 5—Extremely bothered).

**Table 3 ijerph-18-12710-t003:** Association between the level of Perceived Burden of Therapy (PBOT) and patient demographics.

Characteristics	Low Level of PBOT N (%) (n = 45)	High Level of PBOT N (%) (n = 8)	*p* Value
Age (years) *	59.1 ± 15.0	44.5 ± 13.5	0.02
Gender			0.34
Male	16 (35.6)	4 (50)	
Female	29 (64.4)	4 (50)	
Phosphate binder ^¥^			0.57
Yes	41 (91.1)	7 (87.5)	
No	4 (8.9)	1 (12.5)	
Folic acid intake ^¥^			0.25
Yes	13 (28.9)	4 (50)	
No	32 (71.1)	4 (50)	
MCV *	86.8 ± 7.1	88.6 ± 7.0	0.46
Number of medications *	11.0 ± 3.1	11.0 ± 1.51	0.90
Number of pills *	(16 ± 3.6)	(14.4 ± 5.8)	0.28
Number of comorbidities *	4.7 ± 2.3	4.1 ± 1.7	0.44

* (mean ± SD; Mann—Whitney test); ^¥^ (Chi-square test or Fisher’s exact test). Perceived Burden of Therapy (PBOT), Mean Corpuscular Volume (MCV).

## Data Availability

The datasets generated and analyzed during the current study are not publicly available due to participant confidentiality but are available from the corresponding author on reasonable request.

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
