# Peer review of "Assessment of Dietary Folate Intake and Pill Burden among Saudi Patients on Maintenance Hemodialysis"

_ijerph, 2021, doi:10.3390/ijerph182312710_

Round 1

Reviewer 1 Report

-In this manuscript, the authors studied about “Assessment of Dietary Folic Acid Intake and Burden of Oral Therapy among Saudi Patients on Maintenance Hemodialysis: A Cross-Sectional Single-Center Study”. The Authors reported that folic acid intake in among Saudi Patients. This paper can be interesting but the Authors should explain in more detail points and verify their results with more figures data. However, there are some concerns before its rejection. These points are valid in this manuscript rejections.  

-Comments:

Q1: Title revision is required; it is not effective and too long.

Q2: I feel, the materials and methods part are too length and not divided as standard format.  Materials and methods part is written as a passage. This is very hard to follow the study.

Q3: In this manuscript, lot of space grammatical mistake is found.

Q4: Conclusion part should hold those important results outcome and significance from this research analysis. This part is not effective.    

Q5: All table could be improved with better quality. Make sure on all number and space in all over table. No abbreviation is found in below all tables.  

Q6: In Figure 1, table 1 to 3 and their caption is not impressive. Explain it and improve figure and table quality.

Q7: Ref. 13, 16, 20, 22, 25 to 27, Please make a reference alignment as uniformly as per MDPI template. I feel, MDPI required all authors list before title (not: single author et al.,). Please refer for further information at Int. J. Environ. Res. Public Health 202118(20), 10873; https://doi.org/10.3390/ijerph182010873.

Reviewer 2 Report

There were nothing new in this manuscript.  

Reviewer 3 Report

I looked at the manuscript of Sales et al. entitled “Assessment of Dietary Folic Acid Intake and Burden of Oral 2 Therapy among Saudi Patients on Maintenance Hemodialysis: 3 A Cross-Sectional Single-Center Study.” They performed a cross-sectional study in Saudi patients on maintenance hemodialysis. They found that folate supplementation in Saudi patients on maintenance hemodialysis may not be necessary due to their adequate dietary folate intake. However, a cross-sectional study sample size in Saudi patients on maintenance hemodialysis was very small over three months. Therefore, it is hard to agree that the amount of folate intake in Saudi patients on maintenance hemodialysis may reflect the whole population of Saudi patients. But, they found that the omission of folate supplementation may 36 be advantageous for the particular population plagued with pill burden among Saudi patients on Maintenance Hemodialysis. 

Minor points.

There are so many spaces errors after references. Would you please correct those? 

Reviewer 4 Report

The topic is of interest to the academic and scientific community, but requires some changes:

PURPOSE - The objective in the ABSTRACT is different from the one in the text, at the end of the Introduction, in lines 69-70.
MATERIALS AND METHODS - Greater clarity is needed in data collection in relation to the place, time and form of approach. The authors stated that they collected the data through structured "Interviews", carried out independently. But on page 2, line 89, the authors refer to a "Questionnaire" to collect food frequency data. It is necessary to remember that an Interview is different from a Questionnaire.
RESULTS - On page 3, lines 114-116, the authors mention the number of hemodialysis sessions that patients performed per week. It must be made clear that as this is a cross-sectional study, at which point the patients participated in the research.
DISCUSSION - The authors discuss well the data regarding dietary intake of folate among Saudi patients undergoing hemodialysis and adequately refer to the discomfort with the amount of pills taken, adverse effects and medication adherence. These considerations contemplate the objective of the study, but they are not included in the conclusions of the article.
CONCLUSIONS - They do not meet the objective of the research. They do not refer to folate intake, nor do they talk about the impact of oral therapy/medication compliance.
It is noteworthy that they should avoid the term "impact" on the objective, because it
it always requires the consideration of the Before and After of a given phenomenon analyzed.

The topic is of interest to the academic and scientific community, but requires some changes:
PURPOSE - The objective in the ABSTRACT is different from the one in the text, at the end of the Introduction, in lines 69-70.
MATERIALS AND METHODS - Greater clarity is needed in data collection in relation to the place, time and form of approach. The authors stated that they collected the data through structured "Interviews", carried out independently. But on page 2, line 89, the authors refer to a "Questionnaire" to collect food frequency data. It is necessary to remember that an Interview is different from a Questionnaire.
RESULTS - On page 3, lines 114-116, the authors mention the number of hemodialysis sessions that patients performed per week. It must be made clear that as this is a cross-sectional study, at which point the patients participated in the research.
DISCUSSION - The authors discuss well the data regarding dietary intake of folate among Saudi patients undergoing hemodialysis and adequately refer to the discomfort with the amount of pills taken, adverse effects and medication adherence. These considerations contemplate the objective of the study, but they are not included in the conclusions of the article.
CONCLUSIONS - They do not meet the objective of the research. They do not refer to folate ingestion, nor do they talk about the impact of oral therapy/medication compliance.
It is noteworthy that they should avoid the term "impact" on the objective, because it
it always requires the consideration of the Before and After of a given phenomenon analyzed.

Round 2

Reviewer 1 Report

Nothing 

Author Response

Dear reviewer, many thanks for all suggestions and comments, which helps to improve our manuscript, we appreciate all efforts of your team. 

Reviewer 2 Report

This manuscript was corrected according to my comments.

Author Response

We appreciate the reviewer in all stages of review process and many thanks 

Reviewer 4 Report

Dear authors,

The text has improved a lot. However, in the MATERIAL AND METHODS item, there is still a need for greater clarity regarding data collection.  I ask: Which data collection instrument was used in the two collection moments? Was it an INTERVIEW in both moments? Or was it an interview in the first moment and a QUESTIONNAIRE in the second moment? Please remember that Interview, Questionnaire and Forms are different!

Another aspect: In the text there are phrase ou paragraphs repeated in different places. Examples:

a) In Materials and Methods, item 2.2 "Participants", the phrase "Patients were excluded if they were on peritoneal dialysis, were pregnant, had psychiatric illnesses ou were under palliative care" is repeated in item 2.6 Ethical Considerations. I suggest leaving this sentence only in item 2.2 "Participants",  because it is about the Exclusion Criteria of the participants. 

b) In item 2.4, Data Collection, the phrase "During this phase, researchers observed each other to standardize the interviewing practice and minimize variations in data collection" , is repeated in Ethical Considerations. I suggest leaving this sentence in item 2.4 Data  Collection, because these are Data Collection procedures.

c) In item 2.4 Data collection, when referring to the length of time of 20-30 minutes of collection, I suggest that instead of writing that the DATA COLLECTION lasted approximately.... refer to the instrument used (Interview or Questionnaire) and not data collection. Justification= When we refer to the time of data collection, we generally are referring to the entire survey collection period, in this case, 03 months.

Another consideration= As the article  has been translated,  there may be differences in the meaning of sentences or paragraphs . I suggest  the authors a final review of the English language.

Author Response

The text has improved a lot. However, in the MATERIAL AND METHODS item, there is still a need for greater clarity regarding data collection.  I ask: Which data collection instrument was used in the two collection moments? Was it an INTERVIEW in both moments? Or was it an interview in the first moment and a QUESTIONNAIRE in the second moment? Please remember that Interview, Questionnaire and Forms are different!

There are phrases ou paragraphs repeated in different places. Examples:

  1. a) In Materials and Methods, item 2.2 "Participants", the phrase "Patients were excluded if they were on peritoneal dialysis, were pregnant, had psychiatric illnesses ou were under palliative care" is repeated in item 2.6 Ethical Considerations. I suggest leaving this sentence only in item 2.2 "Participants",  because it is about the Exclusion Criteria of the participants. 
  2. b) In item 2.4, Data Collection, the phrase "During this phase, researchers observed each other to standardize the interviewing practice and minimize variations in data collection" , is repeated in Ethical Considerations. I suggest leaving this sentence in item 2.4 Data  Collection, because these are Data Collection procedures.

Thank you for your comment. We are sorry for the confusion. The clean version of the updated manuscript did not have either repetition under the Ethical considerations section. The Ethical considerations section simply states that: “This study was approved by the KSUMC Institutional Review Board Project No. E-19-3858.”

Maybe there was a problem with the track changes function.

  1. c) In item 2.4 Data collection, when referring to the length of time of 20-30 minutes of collection, I suggest that instead of writing that the DATA COLLECTION lasted approximately.... refer to the instrument used (Interview or Questionnaire) and not data collection. Justification= When we refer to the time of data collection, we generally are referring to the entire survey collection period, in this case, 03 months.

Thank you for your suggestion. The sentence was modified and now states, “Both questionnaires were administered during a single MHD session, and the researchers spent approximately 20-30 minutes with each participant.”

Another consideration= As the article has been translated, there may be differences in the meaning of sentences or paragraphs. I suggest the authors a final review of the English language.

Thank you for your comment. The entire manuscript was English edited.